

# Validity and reliability evidence of a point of care assessment of salivary cortisol and α-amylase: a pre-registered study

Kagan J. Ducker[1], Robin L.J. Lines[1], Michael T. Chapman[1], Peter Peeling[2], Alannah K.A. McKay[2,3] and Daniel F. Gucciardi[1]

[1] School of Physiotherapy and Exercise Science, Curtin University, Perth, Western Australia, Australia
[2] School of Human Sciences (Exercise and Sport Science), The University of Western Australia, Perth, Western Australia, Australia
[3] Australian Institute of Sport, Belconnen, Australian Capital Territory, Australia

Corresponding author
Daniel F. Gucciardi,
daniel.f.gucciardi@gmail.com

## ABSTRACT

**Purpose.** The iPro Cube is a small portable point-of-care device designed to analyse salivary markers of stress in a user-friendly manner (e.g., fast, convenient). Our aim was to test the reliability and validity of the iPro Cube to measure salivary cortisol and $\alpha$-amylase as compared to the common laboratory standard method (ELISA immunoassay) prior to and after moderate intensity exercise.

**Methods.** The study was a repeated measures, pre-registered design, and statistical framework that incorporated prior knowledge directly into the estimation process. Twenty-nine individuals (age $= 27.4 \pm 6.6$ y; body-mass $= 70.8 \pm 11.3$ kg; height $= 1.74 \pm 0.92$ m; 18 males) completed a single $PWC_{75\%HRmax}$, with repeated measures of salivary cortisol and ά-amylase pre, immediately post, and 30 min post-exercise.

**Results.** Correlation between the iPro Cube and laboratory-based assessments of salivary cortisol was moderate-to-large ($0.53 > r < 0.81$) across all three testing points. In contrast, correlation between the iPro Cube and laboratory-based assessments of ά-amylase was small-to-moderate ($0.25 > r < 0.46$). We found a large correlation between duplicate samples of iPro Cube cortisol assessment ($0.75 > r < 0.82$), and a moderate-to-large correlation for ά-amylase ($0.51 > r < 0.77$).

**Conclusions.** The iPro Cube is capable of taking measures of salivary cortisol that are moderately correlated to values obtained via ELISA immunoassay, however the unit underestimates salivary cortisol and overestimates salivary ά-amylase at rest and post-moderate intensity exercise. It is recommended that researchers continue using standard laboratory techniques to assess these salivary stress markers.

## INTRODUCTION

The stress placed on the body by exercise can be measured using various biological markers in saliva, hair, and blood. Two key systems in the neuroendocrine response to stress are the hypothalamic-pituitary-adrenal (HPA) axis and the autonomic nervous

system (ANS), which drive responses to assist the body in coping with the applied stressor (*Rutherford-Markwick et al., 2017*; *Strahler et al., 2017*). Salivary markers of HPA axis and ANS activity have increased in popularity recently, as there are several metabolites that can be detected via this non-invasive technique, using only small sample volumes (i.e., ∼0.5–1 ml) (*Lee, Kim & Choi, 2015*; *Papacosta & Nassis, 2011*). Collection of salivary metabolites offers a viable non-invasive alternative to serum and plasma methods where frequent venepuncture to sample blood may impact the analyte of interest (e.g., increasing cortisol response) or may be inappropriate, requiring less medical expertise, rapid collection, and can be performed in a variety of settings. Salivary cortisol (HPA axis) and alpha-amylase ($\alpha$-amylase; ANS) are two such markers that can be measured in the saliva as a surrogate measure of psychophysiological stress.

Cortisol is a glucocorticoid released from the adrenal cortex by activation of the HPA axis when the body is stressed (*Lee, Kim & Choi, 2015*). Only a thin layer of cells separate the salivary ducts from blood passing through the area, which ultimately results in trace levels of cortisol passing through the oral cavity into saliva; when measured, salivary cortisol typically provides a strong representation of plasma cortisol levels (*Törnhage, 2009*), though disassociations have been observed under challenge situations, during the circadian cycle, and in terms of free versus protein-bound cortisol concentrations (*Levine et al., 2007*) . Salivary cortisol is advocated for use within stress research because of the generally strong representation of plasma cortisol levels and stress free nature of collection (*Hellhammer, Wüst & Kudielka, 2009*). Cortisol has widespread effects (e.g., stimulation of protein and lipid catabolism, increased gluconeogenesis and reduced inflammation) that are aimed at enhancing the ability of the body to cope with stress and repair tissue damage (*Lee, Kim & Choi, 2015*; *Levine et al., 2007*). Importantly, when the HPA axis is chronically stimulated, the typical responsiveness of the axis may be supressed, which may inhibit immune function (*Adam et al., 2017*). The responsiveness of the HPA axis makes cortisol a useful marker for assessing the acute (short-term) and chronic (long-term) response to stress in individuals and athletes.

Alternatively, $\alpha$-amylase is an enzyme that is secreted directly into the saliva in response to ANS activation, as would commonly occur when humans experience psychophysiological stress (Bosch, Veerman, de Geus, & Proctor, 2011; (*Nater & Rohleder, 2009*). Importantly, the salivary concentration has a consistent circadian rhythm and is highly responsive to changes in ANS activity as it is produced within the salivary glands rather than being transported through into the salivary ducts from plasma like cortisol (*Nater & Rohleder, 2009*; *Rohleder & Nater, 2009*). During exercise, and particularly during high-intensity efforts, the ANS is activated, thereby making $\alpha$-amylase a potentially useful indicator of physiological stress in humans. Caution is advised when using $\alpha$-amylase as a measure of ANS activity (*Bosch et al., 2011*), as stress induced increases in $\alpha$-amylase have been found to correlate poorly with other indicators of ANS activity (e.g., catecholamines) (*Thoma et al., 2012*). Nevertheless, this poor degree of association is a common issue with indicators of ANS activity, and in turn is not considered a disqualifying factor (*Engeland, Bosch & Rohleder, 2019*).

[1] *iPro Lab reader*—this is iPro's larger, and older, saliva analysis system. It reads the lateral flow device (LFD) that contains the samples of saliva. See http://somabioscience.com/?page_id=214. We will simply refer to this as the iPro Lab.

[2] *iPro Cube reader* - this is iPro's smaller, and newer, portable saliva analysis system. It reads the lateral flow device (LFD) that contain the samples of saliva. See http://somabioscience.com/?page_id=182. We will simply refer to this as the iPro Cube.

[3] *iPro Cube LFD* – We will simply refer to this as a lateral flow device (LFD). The LFD's are the strips that contain the saliva samples and are inserted into either the iPro Lab or Cube devices to analyse the saliva samples.

Typically, stress markers such as cortisol and $\alpha$-amylase are assessed via a sample of saliva or blood, which are analysed in a laboratory by highly skilled technicians using processes that are expensive and can take several hours or days. However, small portable point-of-care (POC) devices are available to make this process easier to conduct in the field,

with analysis requiring short timeframes that allow the rapid generation of information to assess the stress response rapidly. Recently, Soma Bioscience (Oxfordshire, UK) have released two lateral flow device (LFD) readers, the iPro Lab[1] and iPro Cube[2] . Immunochromagraphic strips (lateral flow devices—LFD [3] ) containing a test line that responds to gold labelled anti-secretory antibodies are exposed to a saliva sample (and buffer) which can be assessed by the reader to obtain a measure of salivary cortisol and $\alpha$-amylase. The iPro Lab has been shown to be a valid and reliable unit to assess salivary cortisol at rest (*Fisher, McLellan & Sinclair, 2015*) and following high-intensity exercise (sprinting) (*MacDonald, Bellinger & Minahan, 2017*). The iPro Lab's utility in measuring salivary cortisol has been validated against traditional methods, demonstrating non-significant differences in resting cortisol levels with the Salimetrics oral swab ($p = 0.881$) and the passive drool method ($p = 0.145$) (*Fisher, McLellan & Sinclair, 2015*). Evidence also advocates its use in assessing cortisol levels following high-intensity exercise (sprinting), demonstrating significant increases from resting to 15 min post exercise (*MacDonald, Bellinger & Minahan, 2017*). Caution is urged when interpreting these findings as the sample sizes for the chosen designs were small. The iPro Lab has also been validated at assessing the immune marker, salivary immunoglobulin-A, at rest (*Coad, Gray & McLellan, 2016*; *Coad et al., 2015a*; *Coad et al., 2015b*), and after team-sport (intermittent) exercise (*Coad, Gray & McLellan, 2016*; *Coad et al., 2015b*). However, whether the newer and more portable version of this device (iPro Cube) can be used to assess $\alpha$-amylase at rest, and to measure relatively small changes in salivary cortisol and $\alpha$-amylase post-moderate intensity exercise remains unknown. This information is important for researchers, sports scientists, and health professionals who are seeking a POC device to assess markers of psychophysiological stress in a field setting. Therefore, our aim was to test in the field the reliability and validity of the iPro Cube to measure salivary cortisol and $\alpha$-amylase as compared to the common laboratory standard method (ELISA). On the basis of a conservative estimate from past research (*Coad et al., 2015a*; *Coad et al., 2015b*; *Fisher, McLellan & Sinclair, 2015*), we hypothesised that;

1. For salivary cortisol and $\alpha$-amylase, respectively, POC assessment will correlate positively and moderately ($r \sim 0.50$) with a laboratory-based assessment (a) pre-exercise (resting baseline), (b) immediately post-exercise, and (c) 30 min post-exercise.

2. Duplicate samples collected concurrently of the POC assessment of salivary cortisol and $\alpha$-amylase, respectively, will correlate positively and strongly ($r \sim 0.80$).

## MATERIALS AND METHODS

### Participants

Monte Carlo simulations (L. K. (*Muthén & Muthén, 2002*) indicated that 15 participants would provide sufficient power (100%) to detect a conservative effect size ($r \sim 0.50$) when incorporating prior beliefs informed by past research (*Coad et al., 2015a*; *Coad et al., 2015b*; *Fisher, McLellan & Sinclair, 2015*) directly into the estimation process. Full details of these power simulations are available on the Open Science Framework (OSF; http://bit.ly/2HCA6MR). Individuals were eligible to participate if they were university students aged $\geq 18$ years; we excluded people from taking part in the study if they had an illness or injury that prevented them from completing a sub-maximal exercise test. In total, 29 eligible individuals provided written informed consent and took part in this study (age = $27.4 \pm 6.6$ y, range = 19–48 y; body-mass = $70.8 \pm 11.3$ kg; height = $1.74 \pm 0.92$ m; BMI = $23 \pm 3$; 18 males). The study was approved by Curtin University's Human Ethics Research Committee (HRE2017-0518).

### Procedures

All study hypotheses, methods, procedures and data analysis plans were pre-registered on the OSF (https://osf.io/tshkc/). The study design was a single-session exercise protocol with repeated measures of salivary cortisol and $\alpha$-amylase prior to completing a sub-maximal test of aerobic work capacity, immediately following this test, and after 30 min of rest. Each participant attended one testing session (08:00–10:00 or 10:00–12:00) to minimise any effect from circadian variations. Waking time was not recorded, however, there should have been >60 min between the participants waking to the start of salivary sampling, to ensure that the cortisol awakening response was avoided. Upon arriving in the laboratory and providing informed consent, participants completed a short survey including self-reported demographic information (e.g., gender and age). Participants next completed a sub-maximal test of aerobic work capacity (PWC$_{75\%HRmax}$; (*Gore et al., 1999*; *Miyashita et al., 1985*) where there would be some stimulation of the HPA. Briefly, this test involves completing three workloads of 3–6 min at approximately 55, 65 and 75% of age predicted maximum heart rate (220 bpm –age in years) on a cycle ergometer (Monark, Monark Exercise AB, Varnsbro, Sweden), which was used as a method of prescribing a bout of individualised moderate intensity physical activity. This validation study was part of a larger project our team was conducting with Army personnel, so we decided to focus on an intensity that was most representative of typical training scenarios (*Friedl et al., 2012*). Previous studies have reported increases in both salivary cortisol and $\alpha$-amylase in response to a single bout of exercise (*Strahler et al., 2017*). Heart rate and ratings of perceived exertion (*Borg, 1982*) were collected each minute during the test.

Participants were required to avoid high-intensity exercise and alcohol for 24 h and brushing their teeth or eating food for 2 h pre-exercise. Samples of saliva were taken pre-warm-up, immediately post-exercise and 30 min post-exercise. Participants rinsed their mouth with water 10 min prior to saliva collection at pre-warm-up. Nine unstimulated whole saliva samples were provided by each participant ($3 \times$ pre-warm-up, $3 \times$ immediately post-exercise and $3 \times$ 30 min post-exercise). On each occasion

duplicate oral swabs (iPro oral fluid collector; Soma Bioscience, Oxfordshire, UK) were placed in the mouth concurrently for assessment using a cortisol/ $\alpha$-amylase LFD before being analysed using a LFD reader (iPro Cube, Soma Bioscience, Oxfordshire, UK). The intra-sample coefficient of variation (CV) for the iPro salivary cortisol $\alpha$-amylase samples were 8.8 and 20.0%, respectively. Next, the third sample was collected by the passive drool method directly into a two mL cryovial immediately after the sample for the iPro Cube (Salimetrics LLC, Pennsylvania, USA). This sample was frozen at $-20\,°C$ until analysis in duplicate for cortisol and $\alpha$-amylase using a validated, commercially available enzyme-linked immunosorbent assay (ELISA; Salimetrics, USA) by Stratech Scientific APAC Pty Ltd. (Sydney, Australia). Freezing has minimal influence of samples at $-20\,°C$ even after one year in storage (*Garde & Hansen, 2005*). The inter-assay and intra-assay CV for the salivary cortisol samples was 5.0 and 4.4%, respectively. The inter-assay and intra-assay CV for the salivary $\alpha$-amylase samples was 5.8 and 5.1%, respectively.

## Statistical analyses

The registered study hypotheses were tested in M*plus* 8 (L. K. (*Muthén & Muthén, 1998-2017*)-2017) using a series of bivariate correlations with a Bayesian estimator. The execution and reporting of these analyses were informed by recent guidelines for Bayesian statistics (*Depaoli & Van de Schoot, 2017*). An overview of the priors for each hypothesis is available on the OSF. We also report and compare our weakly or informative priors with non-informative (diffuse) priors to examine the effect of prior information on the posterior distribution (*Depaoli & Van de Schoot, 2017*). Briefly, the specification of non-informative priors allows the data at hand to drive the estimations within the posterior distribution because it reflects substantial uncertainty regarding the nature of the target parameter. Model convergence was assessed using statistical (i.e., potential scale reduction factor; PSR <1.05) and visual criteria (i.e., inspection of trace plots for stability in mean and variance of each chain). Posterior predictive checking is used to assess model fit in Bayesian estimation, where the posterior distribution is compared with the observed data to examine the degree to which the replicated data matches the observed data (B. O. (*Muthén & Asparouhov, 2012*). The posterior predictive *p*-value (PPP) and associated 95% credibility interval (CI) is produced in M*plus*; values close to 0.50 reflect an excellent fitting model, though typically values greater than 0.05 are considered acceptable (B. O. (*Muthén & Asparouhov, 2012*). Parameter estimates are considered credible when the 95% CI excludes zero. We classified "not a detectable" (NAN) readings as a missing value; missing data were handled with the Gibbs sampler that treats the missing observations as unknown values to be estimated and the algorithm used will correctly estimate the model under the missing at random (MAR) assumption (*Asparouhov & Muthén, 2010*). All M*plus* output files including analysis syntax together with Bland-Altman plots (*Bland & Altman, 1986*) are available on the OSF (https://osf.io/tshkc/).

## RESULTS

The raw data file is available on the OSF. Descriptive statistics for the heart rate and perceived exertion response to the exercise stimulus are presented in Table 1. In total,

**Table 1 Descriptive statistics for heart rate, percentage of age predicted heart rate maximum ($\%HR_{max}$ = 220 bpm–age in years), and rating of perceived exertion (RPE 6–20; (*Borg, 1982*) during workloads 1–3 of the $PWC_{75\%HRmax}$ test.**

| | | Range | | Mean | SD |
|---|---|---|---|---|---|
| | | Minimum | Maximum | | |
| Heart rate (bpm) | Workload 1 | 86 | 128 | 110 | 8 |
| | Workload 2 | 97 | 158 | 129 | 11 |
| | Workload 3 | 124 | 170 | 149 | 8 |
| $\%HR_{max}$ | Workload 1 | 50% | 64% | 56% | 3% |
| | Workload 2 | 53% | 81% | 66% | 5% |
| | Workload 3 | 70% | 87% | 77% | 4% |
| RPE | Workload 1 | 6 | 14 | 10 | 2 |
| | Workload 2 | 6 | 17 | 12 | 3 |
| | Workload 3 | 7 | 19 | 14 | 3 |

**Notes.**

bpm, beats per minute; $\%HR_{max}$, Percentage of maximum heart rate; REP, Rate of perceived exertion.

49 (47 of which were obtained with the iPro Cube) of 522 data points were recoded as missing due to an undetectable reading; 24 of the iPro Cube samples were recoded as 0 when the device provide a value that was <0.7.

Group-level descriptive statistics for salivary cortisol and $\alpha$-amylase are presented in Table 2. Analysis of the Bland-Altman plots (available on the OSF; https://osf.io/tshkc/) show a positive mean difference in cortisol concentrations at all time points of the assessment, indicating that the iPro Cube underestimates salivary cortisol. Additionally, the difference between the iPro Cube and immunoassay increases with greater cortisol concentrations. A negative mean difference in $\alpha$-amylase was also observed, indicating that the iPro Cube overestimates $\alpha$-amaylase. Further, this difference becomes greater with increasing levels of $\alpha$-amaylase.

With regard to the main analyses, the probability of the data, given the hypothesised model, was acceptable for all models associated with each hypothesis (i.e., PPP >0.20). Inspections of the trace plots and PSR evolution through the simulations (i.e., started within 0.50 of 1 and reduced quickly close to 1 and remained stable throughout the entire sequence of iterations) verified support for model convergence. Full details of model-data fit and convergence are available in the output and supplementary material files located on the OSF. Parameter estimates of all models tested using weakly informative (uniform), informative, and non-informative (diffuse) priors are displayed in Table 3. We also report in Table 3 parameter estimates obtained using a robust maximum likelihood estimator for readers most familiar with frequentist statistics.

Overall, these analyses revealed mixed support for our hypotheses. Correlation between POC and laboratory-based assessments of salivary cortisol was moderate-to-large ($0.53 > r < 0.81$) across all three testing points. In contrast, correlation between POC and laboratory-based assessments of $\alpha$-amylase was small-to-moderate in magnitude ($0.25 > r < 0.46$). A similar trend in the findings was observed for hypothesis 2, where we found large correlation between duplicate samples of POC assessment of cortisol

**Table 2** Descriptive statistics for salivary cortisol and a-amylase pre-warm-up, immediately post- exercise and 30 min post-exercise.

| | | | | Range | | | |
|---|---|---|---|---|---|---|---|
| | | | *n* | Minimum | Maximum | Mean | SD |
| Salivary cortisol (nMol L$^{-1}$) | Pre-exercise | iPro Cube 1 | 29 | 0.00 | 15.10 | 3.54 | 3.40 |
| | | iPro Cube 2 | 21 | 0.00 | 8.80 | 2.70 | 2.32 |
| | | ELISA | 29 | 2.21 | 28.14 | 6.92 | 2.96 |
| | Post-exercise | iPro Cube 1 | 27 | 0.00 | 9.60 | 2.28 | 2.19 |
| | | iPro Cube 2 | 24 | 0.00 | 6.30 | 2.09 | 1.76 |
| | | ELISA | 28 | 2.21 | 13.79 | 6.18 | 3.21 |
| | 30 min post-exercise | iPro Cube 1 | 25 | 0.00 | 5.80 | 2.23 | 1.62 |
| | | iPro Cube 2 | 23 | 0.00 | 7.20 | 2.27 | 2.10 |
| | | ELISA | 29 | 1.10 | 8.55 | 4.64 | 2.08 |
| Salivary α-amylase (nKat L$^{-1}$) | Pre-exercise | iPro Cube 1 | 28 | 0.50 | 5.25 | 2.00 | 1.30 |
| | | iPro Cube 2 | 23 | 0.48 | 14.58 | 3.43 | 4.15 |
| | | ELISA | 29 | 0.12 | 4.22 | 0.99 | 0.80 |
| | Post-exercise | iPro Cube 1 | 28 | 0.28 | 14.33 | 3.49 | 3.40 |
| | | iPro Cube 2 | 22 | 0.32 | 10.98 | 3.35 | 3.12 |
| | | ELISA | 28 | 0.11 | 3.62 | 1.32 | 0.93 |
| | 30 min post-exercise | iPro Cube 1 | 29 | 0.27 | 10.52 | 2.47 | 2.32 |
| | | iPro Cube 2 | 22 | 0.27 | 8.87 | 1.86 | 1.81 |
| | | ELISA | 29 | 0.09 | 2.53 | 1.00 | 0.65 |

($0.75 > r < 0.82$), but moderate-to-large correlation $\alpha$-amylase ($0.51 > r < 0.77$). A closer inspection of the findings revealed variation in the substantive interpretation of the results in two ways. First, the strength of association between the POC assessment and laboratory analysis of salivary cortisol and $\alpha$-amylase varied across the duplicate samples at each assessment point (24–50%). Second, the discrepancies between the weakly informative and informative priors with the non-informative prior indicated moderate-to-large effects of the prior information on the posterior distribution, which represents a mismatch between prior expectation and the data at hand (*Depaoli & Van de Schoot, 2017*).

## DISCUSSION

We tested the reliability and validity of the iPro Cube as a rapid field assessment of salivary cortisol and $\alpha$-amylase, comparing the outcomes to the most common laboratory method (ELISA) utilising a pre-registered design and statistical framework that incorporated prior knowledge directly into the estimation process. Our findings revealed mixed support for the hypotheses.

During the exercise testing, participants experienced peak heart rates reflective of moderate- to vigorous-intensity exercise (*Norton, Norton & Sadgrove, 2010*), and RPE that were indicative of light to somewhat hard efforts (*Borg, 1982*) (see Table 1). Considering that this test was graded, and only involved short workloads, the overall physical stress placed on the participants was moderate, but only for a short period. Although the response of salivary cortisol to exercise can be variable, previous research

**Table 3 Comparison of standardised correlation coefficients between Bayesian and maximum likelihood estimators.** The two correlation coefficients presented for each variable in the ELISA column are those between the ELISA and each of the two iPro Cube samples. The single value for each variable in the iPro Cube column is the correlation coefficient for the two iPro Cube samples; 95% credibility intervals are presented in parentheses; MLR = robust maximum likelihood estimator; ELISA = analysis via ELISA immunoassay; iPro be = correlation between duplicate samples via the point of care assessment; # SOE (size of effect) = [(initial prior / default/non-informative prior)/ initial prior] * 100 (see *Depaoli & Van de Schoot, 2017*).

| | | Bayesian Uniform/Weakly Informative Prior | | Bayesian Non-informative Prior | | SOE# | | MLR | |
|---|---|---|---|---|---|---|---|---|---|
| | | ELISA | iPro Cube | ELISA | iPro Cube | ELISA | iPro Cube | ELISA | iPro Cube |
| Salivary cortisol (nMol L$^{-1}$) | Pre-exercise | 0.75 (0.56, 0.84) | 0.79 (0.64, 0.88) | 0.81 (0.61, 0.91) | 0.81 (0.58, 0.92) | 8.00% | 2.53% | 0.80 (0.58, 1.02) | 0.82 (0.72, 0.91) |
| | | 0.53 (0.21, 0.72) | | 0.58 (0.21, 0.80) | | 9.43% | | 0.59 (0.35, 0.83) | |
| | Post-exercise | 0.54 (0.23, 0.71) | 0.75 (0.60, 0.85) | 0.59 (0.25, 0.80) | 0.77 (0.51, 0.90) | 9.26% | 2.67% | 0.59 (0.30, 0.87) | 0.77 (0.62, 0.92) |
| | | 0.81 (0.62, 0.89) | | 0.87 (0.69, 0.94) | | 7.41% | | 0.87 (0.77, 0.97) | |
| | 30 min post-exercise | 0.53 (0.20, 0.71) | 0.82 (0.67, 0.90) | 0.59 (0.21, 0.81) | 0.84 (0.63, 0.93) | 11.32% | 2.43% | 0.59 (0.32, 0.86) | 0.85 (0.72, 0.98) |
| | | 0.71 (0.47, 0.83) | | 0.78 (0.53, 0.90) | | 9.86% | | 0.78 (0.64, 0.92) | |
| Salivary α-amylase (nKat L$^{-1}$) | Pre-exercise | 0.44 (0.15, 0.65) | 0.51 (0.37, 0.64) | 0.47 (0.10, 0.73) | 0.23 (-0.23, 0.60) | 6.81% | 54.90% | 0.47 (0.16, 0.77) | 0.23 (-0.37, 0.82) |
| | | 0.32 (0.10, 0.56) | | 0.29 (−0.12, 0.62) | | 9.38% | | 0.29 (0.07, 0.52) | |
| | Post-exercise | 0.46 (0.16, 0.66) | 0.50 (0.36, 0.63) | 0.50 (0.13, 0.74) | 0.24 (−0.19, 0.37) | 8.69% | 52.00% | 0.49 (0.26, 0.73) | 0.24 (−0.19, 0.68) |
| | | 0.37 (0.11, 0.62) | | 0.38 (−0.14, 0.72) | | 2.70% | | 0.39 (−0.04, 0.82) | |
| | 30 min post-exercise | 0.33 (0.10, 0.57) | 0.77 (0.62, 0.86) | 0.31 (−0.09, 0.62) | 0.80 (0.55, 0.92) | 6.06% | 3.90% | 0.31 (.003, 0.61) | 0.81 (0.57, 1.05) |
| | | 0.25 (0.09, 0.51) | | 0.13 (−0.33, 0.53) | | 48.00% | | 0.13 (−0.15, 0.40) | |

suggests that moderate- to vigorous-intensity exercise, particularly when prolonged in nature, will stimulate the HPA and cause an increase in salivary cortisol (*Hayes et al., 2015*). However, due to the short duration and low to moderate initial workloads of the PWC $_{75\%HRmax}$, in our participants, a slight trend for a decrease in salivary cortisol may indicate that the stimulus was insufficiently stressful to activate the HPA axis. If the HPA axis was insufficiently stimulated to achieve an increase, the slight decrease may instead be reflective of the typical diurnal variation, particularly in the 10:00 participants, as cortisol decreases rapidly after an initial increase post-awakening (*Clow et al., 2004*). Both *VanBruggen et al. (2011)* and *Jacks et al. (2002)* identified no increase in salivary cortisol during low (40–45% $VO_{2peak}$) or moderate-intensity (60% $VO_{2peak}$) cycling for 30–60 min, but did see increases during high-intensity (75–80% $VO_{2peak}$) cycling. The intense and prolonged (in the case of (*Jacks et al., 2002*) nature of the high-intensity stimulus in these studies is likely responsibly for the increases in salivary cortisol observed.

The utility of the newer iPro Cube for assessing salivary cortisol in the field has yet to be assessed. In the current study, moderate to large correlations were found between the salivary cortisol measures obtained from the iPro Cube and immunoassay. Additionally, a large correlation was identified between duplicate saliva samples analysed using the iPro Cube. However, although the measures may be related, and duplicate samples are similar for the iPro Cube, there was a mean difference of 51.5–64.6% (underestimate) between the iPro Cube and the immunoassay results. As noted in the data file available on the OSF, 47 of 348 possible samples returned an undetectable result from the iPro Cube. Importantly, the immunoassay has a lower reported limit of quantitation (LOQ) for salivary cortisol (ELISA immunoassay = 0.19 nmol $\cdot L^{-1}$ vs. iPro Cube = 0.58 nmol $\cdot L^{-1}$; (*Salimetrics, 2019a*; *Salimetrics, 2019b*; *The Soma IgA/Cortisol Test With the SOMA Cube Reader, 0000*). These findings raise concerns regarding the reliability and validity of the iPro Cube to assess salivary cortisol and $\alpha$-amylase. *Fisher, McLellan & Sinclair (2015)* have previously reported that the iPro Lab and OFC swab collection method provides valid results compared to ELISA when using both passive drool ($r = 0.45$) and Salimetrics oral swab ($r = 0.52$) to collect the sample. Additionally, they identified that the OFC swab and Lab LFD were reliable when assessing duplicate samples (OFC swab duplicates, $p = 0.81$, ICC=0.89; iPro Lab duplicates, $p = 0.98$, ICC=0.85). However, it is important to bear in the mind the sensitivity of their design (small sample) and relatively weak correlations between methods. The reason for the difference in our findings is unclear as the measurement and analysis techniques were similar, however, the contrast may be related to issues with the manufacturer supplied buffer solution, LFD's, or differences between the performance of the two models of iPro LFD readers.

Alpha-amylase concentrations were found to increase post-exercise before returning to baseline 30 min post-exercise in the current study. Previous research has suggested that $\alpha$-amylase is a sensitive marker of ANS activation as it is produced locally in the salivary glands rather than being a systemic marker (*Rohleder & Nater, 2009*). Our results support the literature in suggesting that $\alpha$-amylase is sensitive to changes in stress induced by exercise, even during short-duration exercise efforts at moderate intensity. For example, *Allgrove et al. (2008)* reported that salivary $\alpha$-amylase concentrations were increased while

cycling at an intensity as low as 50% VO $_{2max}$ for ∼22 min, and were undifferentiated from concentrations recorded at 75% VO $_{2max}$, or exercise to fatigue during a graded exercise test. Similarly, *Kunz et al. (2015)* identified that $\alpha$-amylase was increased during a cycle for 30 min at −5, +5 and +15% of the blood lactate threshold in experienced cyclists, yet the change in concentration was significantly greater in the +15% condition when compared to the −5% condition even though concentrations were similar between trials. With this in mind, future research should investigate the sensitivity of $\alpha$-amylase to detect changes in exercise intensity, particularly during short duration and low to moderate-intensity efforts.

This study is the first to report the efficacy of using either of the iPro LFD readers to assess $\alpha$-amylase at rest and following moderate-intensity exercise. Importantly, it was identified that the validity of the iPro Cube was poor as it was identified that values obtained did not correlate well with the immunoassay (0.50>r<0.77) or between repeated concurrent samples (0.25>r<0.46). Furthermore, the CV within samples was very high for the POC device. Although there are differences in the way that each method determines $\alpha$-amylase, converting both sets of results to enzyme activity using the manufacturer's guidelines should lead to comparable results. The POC unit over-estimated $\alpha$-amylase activity by approximately 150%, which generates concern about the analysis process and/or LFD strips and reader. Early testing for $\alpha$-amylase with the unit encountered consistent "not a number—NAN" error messages, which may relate to the unit measuring what it thought were very high or low concentrations of $\alpha$-amylase. In this instance, the manufacturer checked the buffering solutions and calibration (unique to each unit and buffer solution), which reduced the NAN error messages in subsequent trials, but evidently failed to allow an accurate or consistent assessment of $\alpha$-amylase by the unit. Importantly, the LOQ for the immunoassay is reported (0.58 nKat $\cdot$L$^{-1}$; (*Salimetrics, 2019a*; *Salimetrics, 2019b*), yet we were unable to locate this value for the iPro Cube.

This study had several limitations that should be considered. First, we did not assess the waking time of our participants; although we presume that participants in the first test group (08:00) should have surpassed the peak of the cortisol awakening response, we cannot guarantee that this didn't impact our initial resting measurement in this group. Second, differences in the collection method may explain some variability in the concentrations of cortisol and $\alpha$-amylase. Although the manufacturer has reported that their oral fluid collector recovers >85% of analytes (IgA, cortisol, testosterone and DHEA) in a sample versus passive drool (*Jehanli, Dunbar & Skelhorn, 2011*), this finding is yet to be supported in published, independent research, particularly with $\alpha$-amylase. The assessment of salivary markers is commonly completed using ELISA immunoassay, and immunoassay could potentially be considered the laboratory standard for these assessments, yet liquid-chromatography mass spectroscopy (LC-MSMS) is typically considered the gold standard (*Inder, Dimeski & Russell, 2012*). It is likely these techniques would produce similar results, yet future research should look to compare the measures obtained using the iPro, ELISA and LC-MSMS. Additionally, comparisons between the

more commonly used iPro Lab and new iPro Cube would be beneficial to determine if the units return similar results.

## CONCLUSION

We identified that the iPro Cube reader is capable of reporting measures of salivary cortisol that are moderately correlated to values obtained via the typical laboratory method (ELISA immunoassay). However, the unit was shown to underestimate salivary cortisol and overestimate salivary $\alpha$-amylase at rest and post-moderate intensity exercise when compared to the common laboratory standard ELISA approach. Collectively, therefore, the current findings provide little support for the utility of the iPro Cube as a tool for measuring salivary cortisol and $\alpha$-amylase in the field. As such, researchers, sports scientists, and health professionals are recommended to continue using standard laboratory techniques to assess these salivary stress markers.

### Funding

The Commonwealth of Australia supported this research through the Australian Army and a Defence Science Partnerships agreement of the Defence Science and Technology Group, as part of the Human Performance Research Network (ID7085). The funders had no role in study design, data collection and analysis, decision to publish, or preparation of the manuscript.

### Grant Disclosures

The following grant information was disclosed by the authors:
Australian Army and a Defence Science Partnerships: ID7085.

### Competing Interests

The authors declare there are no competing interests.

### Author Contributions

- Biao Xiong conceived and designed the experiments, performed the experiments, analyzed the data, contributed reagents/materials/analysis tools, prepared figures and/or tables, authored or reviewed drafts of the paper, approved the final draft.
- Limei Zhang analyzed the data, prepared figures and/or tables, approved the final draft.
- Shubin Dong performed the experiments, contributed reagents/materials/analysis tools, prepared figures and/or tables, approved the final draft.
- Zhixiang Zhang conceived and designed the experiments, authored or reviewed drafts of the paper, approved the final draft.

### Human Ethics

The following information was supplied relating to ethical approvals (i.e., approving body and any reference numbers):

The study was approved by Curtin University's Human Ethics Research Committee (HRE2017-0518).

## Data Availability

Supporting documentation (e.g., power simulations, data, analysis scripts) is available on the Open Science Framework: Gucciardi, Daniel, Robin Lines, Michael Chapman, Alannah McKay, and Peter Peeling. 2019. "Assessment of Salivary Cortisol and a-Amylase via Point of Care Assessment (iPro)." OSF. November 4. osf.io/tshkc.

## Supplemental Information

Supplemental information for this article can be found online at http://dx.doi.org/10.7717/peerj.8366#supplemental-information.

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
