# Peer review of "Validity and reliability evidence of a point of care assessment of salivary cortisol and α-amylase: a pre-registered study"

_PeerJ, doi:10.7717/peerj.8366_

## Round 0.1 · original submission · Major Revisions

· Academic Editor

Major Revisions

The reviewers had particular concern with regard to the description of the experimental protocol along with questions about the interpretation of the results in the context of the experimental design. Please address these concerns in your revision. In addition, there seemed to be a lack of clarity that could be improved through reorganization of the manuscript as suggested by the reviewers.

·

Basic reporting

ABSTRACT
Concise and well written
• Line 27 and throughout the manuscript, please be consistent with the terminology to refer to the iPro Lab reader (also appears as Lab LFD reader) and the iPro Cube reader (also appears as iPro Cube, iPro Cube LFD reader and iPro Cube LFD)
INTRODUCTION
Well written
• Line 68 – please review sentence as meaning of “…is advocated within stress…” is unclear
• Line 83 – please review “…through from plasma…”
• Line 88 – suggest to briefly include examples of other indicators for context
• Line 110 – please review grammar “It…”?

Experimental design

MATERIALS AND METHODS
Overview of the relevant components of the registered methodology was provided
• The methodology registered indicated participants were originally University students and subsequently “individuals from the public” however the use of Army personnel is referred to in the text – please clarify the population utilised.
• Lines 152-154 – this justification has little applicability without any reference(s) to support the intensity is indeed representative of Army specific training scenarios. Suggest to readdress the justification of utilising the PWC75%HRmax assessment.
• Line 166 – please review grammar “the iPro salivary cortisol ɑ-amylase samples was”

Validity of the findings

RESULTS
Text was sufficient and Tables are clear and adequate.
• Much of the discussion focusses on under- and over-estimation of samples between iPro Cube LFD reader and ELISA yet no explanation or results of such are presented in the results section. Suggest considering the inclusion of variations in mean differences into Table 1 or the text.

DISCUSSION
• Line 234 – Aren’t the results of the PWC75%HRmax assessment relative to 55%, 66% and 77% rather than low? You also state “moderate-to-vigorous intensity” as well as “light to somewhat hard efforts“. Clarity around the participant composition (such as active, sedentary, military etc) may provide some support to your notion of insufficient stimuli.
• Line 239 – “relatively low-intensity” – see previous comment.
• Line 244 – reference to “the 10:00 participants” is unexplained in either methodology (registered or this manuscript); the methodology in the manuscript states that “All study procedures were completed in one testing session between 8-10 am to minimise any effect from circadian variations.” Please clarify or amend accordingly.
• Line 274 – “relatively moderate intensity” – see previous comments and suggest to revise for consistency throughout the manuscript

CONCLUSION
• Suggest including a conclusive statement regarding the poor validity of the ɑ-amylase

Additional comments

Thank you for the opportunity to review this manuscript which offers an important and practical contribution to the credibility of point of care diagnostics for monitoring and evaluating physiological responses to exercise. The manuscript is well written and provides sufficient overview in its representation of the study methodology, analysis and interpretation albeit differing from the original registered methodology with respect to participants and psychophysiological focus. I agree with the concerns you discuss in your manuscript around the inclusion of the 9 min sub-maximal exercise protocol to elicit meaningful changes in [sCort] which I have mentioned throughout the review. Additionally, the manuscript contains a number of inconsistencies and grammatical errors and with that in mind, I believe some revisions are required for the manuscript and as such, have a few suggestions for your consideration.

Reviewer 2 ·

Basic reporting

This study aimed to investiagte the reliability and validity of iPro Cube LFD, a portable-of-care device, for the immediate assessment of salivary cortisol and alpha-amylase by comparing it to a laboratory standard method, ELISA. Twenty-nine volunteers, undergoing a moderate exercise (3x 3-6 min cycle ergometer at 75%HRmax) in the morning (8-10 am), provided three saliva samples, two swab samples for the iPro and one passive drool sample for the later ELISA analysis (Salimetrics), at three occasions: before, immediately after and 30 min after completion of the exercise. Findings indicate moderate to large correlations for cortisol and low to moderate correlation for alpha-amylase.

This study may contribute to the field by investigating possible reliable and valid alternatives to laboratory methods such as LC-MS/MS and immunoassays. Thus, this study is of most importance and valuable for research and practical implications.

The introduction is somewhat missing conjuctions between and within passages which may cause confusions, misunderstandings and makes is difficult to follow the storyline.
For example, it may seem arbitrary why this study focused on salivary cortisol and alpha-amylase which it is not, of course. You may add a sentence within the first paragraph, naming the two most prominent stress-related biological systems, namely HPA axis and ANS (autonomic nerous system). In doing so, the second and third paragraph dedicated to cortisol and alpha-amylase, respectively, are not disconnected to the previous paragraph anymore.

I have some minor comments concerning the content and cited litertaure.
- Line 57-58: You may add that some analytes may also be affected by the venepucture (for example, see Weckesser et al., 2014 PNEC).
- Line 67: You may add „free versus protein-bound cortisol“. Note, salivary cortisol is only reflecting a fracture of free, unbound, biologically active cortisol.
- Line 69: Please check the reference whether this sentence is true for „free“ plasma cortisol.
- Line 75: Please check the correctness/applicabilty of the reference.
Line 76. What is meant by „chronic response“ and can it really be captured by salivary markers?
- Line 77-89: Although salivary alpha-amylase was originally introduced as an SNS marker, recent reasearch (e.g., Schumacher et al., 2013 PNEC) strongly suggest salivary alpha-amylase as a surrogate for the autonomic nervous system, given that both branches PNS and SNS are involved (e.g., Proctor et al., 2000 Periodontology).
- Line 119/ 122: It is not clear why a moderate and large correlations are assumed (hypothesis 1 and 2, respectively).

Experimental design

In general, I may suggest to structure this section and using subsections/ subheadings:
- Participants: description and recruitment
- Procedure: study design incl. exercise/ prerigisteration information + saliva collection method
- Assessment of cortisol and alpha-amylase: iPro Cube LFD and ELISA
Statistical analyses: statistical procedure

Further comments:
- Line 124-133: Please provide more information about the participants (reference to preregistertaion is not enough, considering that another sample was recruited in the end). Please provide the following information: descriptives such as age (also range), sex, BMI, smoking status (maybe even provide a suppl. table), in- and exclusion criteria beyond the already mentioned ones (e.g., chronic disorders, smoking, medication/ drug intake), and recruitment (apparently, students from the Sports and Exercise Psychology Unit were not recruited?).
- Line 138: Was, by any chance, the time of awakening on the day of assessment was documented, considering the morning awakening response? This might be of help for discussing the non-response in cortisol.
- Line 163: Were saliva samples run in duplicates for the ELISAs?
- Line 171: What data were used for the bivariate correlations - the mean of iPro Cube LFD or the single values of iPro Cube LFD?
- Line 186: How many data were missing in total, and why?

Validity of the findings

The main problem I see is that results and discussion are somewhat all mixed together. For example, in line 224-226 „new“ findings are reported in the discussion.
My suggestion is that you may focus on your main research question but however, I agree that it is important to report the findings of the experiment as well for the interpretation of the results. Please keep in mind, the experiment is not the main research question. Saying that, you may restructure the result section as follows:
- Validation check of the stress induction: biref summary of the findings (later discuss these findings);
- Realibility and validity of iPro Cube LFD: basically the already reported results;

Further comments,
- Line 205-217: I may missed it, would you please provide a table with all correlations.
- Table 1: Apparently, cortisol concentration were not detectable in some of the saliva samples using iPro Cube LFD. How many samples were not detectable? Please discuss this critically.

Another important issue concerns the limitations which should be addressed in a separate section at the end. Some limitations are the following:
- experiment in the morning may overlap with the morning awakening response
- detecable limits of iPro Cube LFD (and ELISAs)
- CV of iPro Cube LFD (also compared to ELISAs) which is also an indicator for reliability in a way
- differences in saliva collection methods that may also partly explain differences and low correlations

Additional comments

All in all, this study provides valuable findings for the field.
At the current state, the manuscript is somewhat missing a clear structure and important information which required to understand the study design and findings. The manuscript may benefit from rewriting and restructuring some passages and addressing the concerns raised (see comments above).

---

## Round 0.2 · Minor Revisions

· Academic Editor

Minor Revisions

Thank you for addressing the reviewer concerns. Reviewer 2 has a few concerns that appear to be relatively minor. Please review these issues and address as appears appropriate.

·

Basic reporting

No comment

Experimental design

No comment

Validity of the findings

No comment

Additional comments

I wish to commend the authors on their comprehensive revisions and as such, believe the manuscript successfully meets the PeerJ criteria and should be accepted. Thank you for the opportunity to review your work.

Reviewer 2 ·

Basic reporting

The authors have addressed all raised concerns adequately. Thank you for clarification and revising the manuscript accordingly.

As to 6, in stress research field and as to my understanding, "response" indicates always a change over at least two time points (pre-post change) and cannot by captured by a single timepoint measure; “chronic stress”, on the other hand, indicates other measures such as urinary or hair cortisol, for example. However, I assume the terms “acute” and “chronic” responses might be used differently, depending on the research field. The addition “short-term” and “long-term” has further contributed to the understanding of what is meant by the two terms.

Please check and correct the spelling of a-amylase in line 223-225 (accidently spelled "a-amaylase").

Experimental design

I appreciate that the authors stucture the manuscript in a reader-friendly way by adding respective subheadings.

As to 8, thank you for introducing salivary alpha-amylase as a surrogate of the ANS.

As to 10, thank you for clarification that the participants were indeed a student sample. Were the students recruited from the Sports and Exercise Psychology Unit only or from various departments?

Again, please provide range in age, BMI (calculated by body mass and height). Thank you.

As to 11, thank you for clarification. The authors have now reported that the assessment started at least 1 hour after awakening which indeed should cover the time period after the morning awakening responses in both biomarkers.

As to 13, you may add the information about how to read the table in the ‘note” under the table as other readers may also ask themselves this question. Thank you.

Validity of the findings

For the interpretation of the findings, I may respond to two authors' repsonses.

As to 15, I see the authors’ point. I agree with the authors that the narrative findings on the exercise performance may not be essential for answering the main research question, as I have stated previously ("... the experiment is not the main research question."). I personally would avoid reporting new findings in the discussion, no matter whether the readers may (or may not) read and interpret the results from the tables by their own. I guess this a question of personal writing style.

As to 16, I may suggest, that the authors may again refer to table 3 in the manuscript. Thank you.

Additional comments

Please accept my apologies for missing some points (comments 4, 12, 18) that have already been addressed in the previous original manuscript. The editiors were desperately in need of a second reviewer and kindly asked me to review the manuscript before the deadline (with the best intention of the editors, namely, to avoid another delay in the already long-lasting peer review process). Thus, I had to review your manuscript under (time) pressure, and I was worried that I may have overlooked some information that have already been reported in the manuscript. This worry has been justified as I see now. This is not an excuse, rather an explanation. Sorry for any inconvenience this may caused. Nonetheless, I hope you could benefit from some of my comments.

---

## Round 0.3 · accepted · Accept

· Academic Editor

Accept

Thank you for addressing the reviewer concerns and congratulations again.